# Binary Classification with Imbalanced Data

**DOI:** 10.3390/e26010015

**Published:** 2023-12-22

**Authors:** Jyun-You Chiang, Yuhlong Lio, Chien-Ya Hsu, Chia-Ling Ho, Tzong-Ru Tsai

**Affiliations:** 1School of Statistics, Southwestern University of Finance and Economics, Chengdu 611130, China; jiangjy@swufe.edu.cn; 2Department of Mathematical Sciences, University of South Dakota, Vermillion, SD 57069, USA; Yuhlong.Lio@usd.edu; 3Department of Statistics, Tamkang University, New Taipei City 251301, Taiwan; 610890161@gms.tku.edu.tw; 4Department of Risk Management and Insurance, Tamkang University, New Taipei City 251301, Taiwan; clho@mail.tku.edu.tw

**Keywords:** artificial neural network, expectation-maximization algorithm, Entropy, logistic regression, zero-inflated model

## Abstract

When the binary response variable contains an excess of zero counts, the data are imbalanced. Imbalanced data cause trouble for binary classification. To simplify the numerical computation to obtain the maximum likelihood estimators of the zero-inflated Bernoulli (ZIBer) model parameters with imbalanced data, an expectation-maximization (EM) algorithm is proposed to derive the maximum likelihood estimates of the model parameters. The logistic regression model links the Bernoulli probabilities with the covariates in the ZIBer model, and the prediction performance among the ZIBer model, LightGBM, and artificial neural network (ANN) procedures is compared by Monte Carlo simulation. The results show that no method can dominate the other methods regarding predictive performance under the imbalanced data. The LightGBM and ZIBer models are more competitive than the ANN model for zero-inflated-imbalanced data sets.

## 1. Introduction

When a data set contains an excess of zero counts over the expectation of a standard statistical distribution, a zero-inflated model helps to analyze such data sets. In many scenarios, count data have two sources of zeros. They are the structural zeros and sampling zeros. Structural zeros mean that the response variable cannot take positive values because of inherent constraints or conditions, and sampling zeros mean the zeros result from random chance. The idea behind the zero-inflated model is to account for two sources of zeros using a mixture of two separated distributions. One model characterizes the probability of structural zeros, and the other describes the non-zero counts. This mixture mechanism of the zero-inflated model allows for better representing the complexities when the data set has excess zeros. The zero-inflated Poisson (ZIP) model and zero-inflated negative Binomial (ZINB) model are two major zero-inflated models.

Imbalanced data refers to the distribution of categories in the data set that is highly disproportionate. For example, in a data set with two types, one type has significantly more instances than another. The situation results in an unequal representation of the two categories. Data sets could be inherently imbalanced in real-world scenarios for various reasons. One example of imbalanced data in medical applications can be diagnosing a rare disease with only a tiny percentage of patients with the condition. In this case, the class size of the rare event will be severely smaller than the common event.

Traditional machine learning algorithms and statistical models could be challenged to handle imbalanced data. The prediction results could be biased toward the majority class and lead to poor performance of the minority class. However, the minority class is often more significantly interesting in many applications. The well-known drawbacks of machine learning algorithms caused by imbalanced data sets are model bias, poor generalization, and misleading evaluation metrics. Several techniques can be used to address the issue of imbalanced data, including resampling, synthetic data generation, using ensemble methods, and more.

The artificial neural network (ANN) has been widely used to implement categorical classification. Rosenblatt [1] proposed a theory of perception about a hypothetical nervous system that can sense, remember, and recognize information from the physical world. Rumelhart et al. [2] discussed a learning algorithm for backpropagation ANNs. The algorithm adjusts the weights of the connections in the network to minimize the error between the realistic output and the desired output. Hirose et al. [3] modified the backpropagation algorithm by varying the number of hidden units in the ANN. Their algorithm can reduce the probability of becoming trapped in local minima and speed up the convergence compared to conventional backpropagation ANN. Sietsma and Dow [4] studied the relationship between the size and structure of an ANN and its ability to generalize from a training set. Using a complex classification task, they investigated the effects of pruning and noise training on network performance.

In the previous decade, Devi et al. [5] emphasized the classification of cervical cancer using ANN methods to develop a strategy for accurately categorizing cervical cancer cases. Sze et al. [6] provided a comprehensive tutorial and survey on the efficient processing of deep neural networks. Abiodun et al. [7] surveyed the ANN applications to provide a taxonomy of ANN models and provided readers with knowledge of current and emerging trends in ANN research. Nasser and Abu-Naser [8] addressed the detection of lung cancer using an ANN method to explore the feasibility of employing ANN technology for identifying lung cancer, which is a significant advancement in medical diagnostics. Muhammad et al. [9] delved into predicting pancreatic cancer using an ANN method. Umar [10] predicted student academic performance using ANN methods to forecast students’ academic achievements. Shen et al. [11] introduced a novel ensemble classification model for imbalanced credit risk evaluation. The model integrates neural networks with a classifier optimization technique. The study addressed the challenge of imbalanced data sets in credit risk assessment by leveraging the strengths of neural networks and advanced optimization techniques.

Lambert [12] proposed a ZIP regression model to process count data with excess zeros. The author applied the ZIP regression model to a soldering defects data set on the printed wiring board and compared the performance of the proposed model with other models. Hall [13] presented a case study of ZIP and zero-inflated binomial (ZIB) regression with random effects. The author showed that zero-inflated models are useful when the data contains an excess of zeros, and random effects can be used to account for the correlation between observations within clusters. Cheung [14] discussed the theoretical underpinnings of zero-inflated models, provided examples of their application, and discussed the implications of using these models for accurate inference in the context of growth and development studies. Gelfand and Citron-Pousty [15] presented how to use zero-inflated models for spatial count data within the environmental and ecological statistics field. They explored how zero-inflated models can account for both the spatial correlation and the excess zero counts often found in environmental and ecological applications. Rodrigues [16] discussed how Bayesian techniques can be applied to estimate the parameters of zero-inflated models and provided practical examples to demonstrate the approach.

Ghosh et al. [17] provided a Bayesian analysis of zero-inflated regression models. Their Bayesian approach suggested a coherent way to handle the challenges of excess zeros in count data, offering insights into parameter estimation and uncertainty quantification. Harri and Zhao [18] introduced the zero-inflated-ordered probit model to handle ordered response data with excess zeros. The proposed method is applied to analyze tobacco consumption data. Loeys et al. [19] explored the analysis of zero-inflated count data, going beyond the scope of ZIP regression. Diop et al. [20] studied the maximum likelihood estimation method in the logistic regression model with a cure fraction. The authors introduced the concept of cure fraction models, which analyze data where many individuals do not experience the event of interest. Staub and Winkelmann [21] studied the consistent estimation of zero-inflated count models. He et al. [22] discussed the concept of structural zeros and their relevance to zero-inflated models. Diop et al. [23] studied simulation-based inference in a zero-inflated Bernoulli regression model. This research is valuable for practitioners working with binary data exhibiting excess zeros.

Alsabti et al. [24] present a novel decision tree classifier called CLOUDS for classification using big data. The authors showed that techniques such as discretization and data set sampling can be used to scale up decision tree classifiers to large data sets. However, both methods can cause a significant accuracy loss. Friedman [25] introduced Gradient Boosting Machines (GBMs) with a powerful technique for building predictive models by iteratively combining weak learners. GBMs are known for their effectiveness in handling complex, high-dimensional data with robustness against overfitting. Jin and Agrawal [26] addressed the challenge of constructing decision trees efficiently in a parallel computing environment. They improved communication and memory efficiency during the parallel construction of decision trees.

Chen and Guestrin [27] developed the “XGBoost”, a scalable tree-boosting system. XGBoost is a powerful and efficient algorithm for boosting decision trees. Ke et al. [28] introduced “LightGBM”, a highly efficient gradient boosting decision tree algorithm. This tool dramatically contributes to machine learning, particularly boosting techniques. LightGBM has gained attention for its speed and scalability, making it suitable for handling large data sets and complex problems. Wang et al. [29] applied “LightGBM” for the miRNA classification in breast cancer patients. Ma et al. [30] used multi-observation and multi-dimensional data cleaning methods and the algorithms of LightGBM and XGboost for prediction. The authors concluded that the LightGBM algorithm based on multiple observational data set classification prediction results is the best. Machado et al. [31] introduced LightGBM, showing it is an excellent decision tree gradient boosting method. The authors applied LightGBM to predict customer loyalty within the finance industry. Minstireanu and Mesnita [32] employed the LightGBM algorithm for online click fraud detection. Daoud [33] compared three popular gradient boosting algorithms, XGBoost, LightGBM, and CatBoost, to evaluate the strengths and weaknesses of each algorithm in the context of credit risk assessment or related tasks.

Some open challenges for implementing imbalanced data analysis have been comprehensively discussed by Krawczyk [34] and Nalepa and Kawulok [35]. The two works provided insights into imbalanced data modeling. Krawczyk [34] pointed out the open challenges including the binary and multi-class classification problems, how to do multi-label and multi-instance learning, and unsupervised and semi-supervised handling for imbalanced data sets. It is also challenging to perform regression on skewed cases and a well-designed procedure for learning from imbalanced data streams under stationary and drifting environments, then extending the model for large-scale and big-data cases. Nalepa and Kawulok [35] reviewed the issue of selecting training sets for support vector machines. They did an extensive survey on existing methods for using the support vector machine method to train data in big-data applications. Nalepa and Kawulok [35] separated these helpful techniques into several categories. This work helps users understand the underlying ideas behind the algorithms used.

Even a ZIBer regression model has been proposed in the literature. How to obtain the maximum likelihood estimates (MLEs) of the ZIBer regression model parameters can be a gap in this topic. We are also interested in studying the performance of the ZIBer regression model compared with machine learning classifiers, for example, the LighGBM and ANN methods. In this study, we propose an EM algorithm to obtain reliable MLEs of the ZIBer regression model parameters based on zero-inflated-imbalanced binary data. The logistic model links the event or non-event probability with explanatory variables. The innovation is to propose the theoretical process of the EM algorithm to obtain reliable MLEs for the ZIBer regression model using imbalanced data sets. Monte Carlo simulations were conducted to show the predictive performance of the ZIBer, LightGBM, and ANN methods for binary classification under imbalanced data.

The rest of this article is organized as follows. Section 2 presents the ZIBer regression model and the proposed EM procedure that produces reliable MLEs of the model parameters. In Section 3, two examples are used to illustrate the applications of using the proposed EM algorithm to the ZIBer regression model. In Section 4, Monte Carlo simulations were conducted to compare the classification rate of the ZIBer model, LighGBM, and ANN methods in three measures. The design of Monte Carlo simulations and their implementation are studied in this section. Some concluding remarks are given in Section 5.

## 2. Zero-Inflated Bernoulli Regression Model and EM Algorithm

Consider a special case about the infection status of a specific disease. $Y_{i}$ denotes the infection status for the specific disease of the *i*-th individual in a sample of size *n*; $Y_{i}=1$ if the individual is infected and $Y_{i}=0$ otherwise. The conditional distribution of $Y_{i}$ can be a Bernoulli distribution, denoted by $Y_{i}|x_{i}∼B\left(1,p_{i}\right)$, where $p_{i}=P\left(Y_{i}=1|x_{i}\right)$ is the conditional probability of $Y_{i}=1$, given a p×1 vector of explanatory variables, $x_{i}^{T}=\left(x_{i1}=1,x_{i2},\;\cdots ,x_{ip}\right)$. The logistic function to link $Y_{i}$ and $x_{i}$ can be expressed by
(1)$$\log\left(\frac{p_{i}}{1-p_{i}}\right)=x_{i}^{T}\beta =\beta_{1}x_{i1}+\beta_{2}x_{i2}+\cdots +\beta_{p}x_{ip},\;i=1,\;2,\;\cdots ,\;n,$$
where $\beta^{T}=\left(\beta_{1},\beta_{2},\;\cdots ,\beta_{p}\right)$ is the coefficient vector. Failure to account for the extra zeros in a zero-inflated model can result in biased estimates and inferences.

Let $\delta_{i}$ control the probability of $Y_{i}$ being a structural zero. $\delta_{i}$ can be an unconditional probability of $Y_{i}$ being a structural zero controlled by an unobserved factor, say $w_{i}$. When $w_{i}=1$, $Y_{i}=0$, otherwise, $w_{i}=0$, $Y_{i}$ can be 0 or 1, determined by the model $B(1,p_{i})$, where $p_{i}=P\left(Y_{i}=1|x_{i}\right))$. It can be shown that the unobserved factor $w_{i}∼B\left(1,\delta_{i}\right)$ for i=1,2,⋯,n. Two situations result in $Y_{i}=0$; the first situation is that $Y_{i}$ is a structural zero with probability $\delta_{i}$, and the other is the situation that $Y_{i}$ is not a structural zero but has the probability $1-p_{i}$ to be zero. The unconditional probability of $Y_{i}=0$ can be presented by
(2)$$P\left(Y_{i}=0\right)=\delta_{i}+\left(1-\delta_{i}\right)\left(1-p_{i}\right)=1-\left(1-\delta_{i}\right)p_{i}=1-\pi_{i},\;i=1,\;2,\;\cdots ,\;n,$$
where $\pi_{i}=\left(1-\delta_{i}\right)p_{i}$. The unconditional probability of $Y_{i}=1$ can be presented by
(3)$$\begin{array}{c} P\left(Y_{i}=1\right)=1-P\left(Y_{i}=0\right)=\pi_{i},\;i=1,\;2,\;\cdots ,\;n. \end{array}$$
We can obtain the unconditional distribution of $Y_{i}$ by $Y_{i}∼B\left(1,\pi_{i}\right)$.

Using the second logistic function to link $\delta_{i}$ and the k×1 covariate vector of $z_{i}^{T}=\left(z_{i1}=1,z_{i2},\;\cdots ,z_{ik}\right)$, we obtain
(4)$$\log\left(\frac{\delta_{i}}{1-\delta_{i}}\right)=z_{i}^{T}\theta =\theta_{1}z_{i1}+\theta_{2}z_{i2}+\cdots +\theta_{p}z_{ik},\;i=1,\;2,\;\cdots ,\;n,$$
where $\theta^{T}=\left(\theta_{1},\theta_{2},\;\cdots ,\theta_{k}\right)$ is the vector of model parameters. After algebraic formulation, we can show that
(5)$$\begin{array}{c} \pi_{i}=\frac{1}{1+e^{z_{i}^{T}\theta }}\times \frac{e^{x_{i}^{T}\beta }}{1+e^{x_{i}^{T}\beta }},\;i=1,\;2,\;\cdots ,\;n. \end{array}$$
Assume that sample $D=(y_{1},\;\cdots ,\;y_{n},\;x_{1},\;\cdots ,\;x_{n},\;z_{1},\;\cdots ,\;z_{n})$ is available. The likelihood function can be constructed as follows:(6)$$\begin{array}{cc} L(\beta ,\theta |D)\; & =\prod_{i=1}^{n}\pi_{i}^{y_{i}}{\left(1-\pi_{i}\right)}^{1-y_{i}}\\ \; & =\prod_{i=1}^{n}{\left(\frac{e^{x_{i}^{T}\beta }}{\left(1+e^{z_{i}^{T}\theta }\right)\left(1+e^{x_{i}^{T}\beta }\right)}\right)}^{y_{i}}\times {\left(\frac{1+e^{z_{i}^{T}\theta }+e^{z_{i}^{T}\theta +x_{i}^{T}\beta }}{\left(1+e^{z_{i}^{T}\theta }\right)\left(1+e^{x_{i}^{T}\beta }\right)}\right)}^{1-y_{i}}. \end{array}$$
After algebraic formulation, we can obtain the log-likelihood equation,
(7)$$\begin{array}{cc} \ell (\beta ,\theta |D)\; & =\sum_{i=1}^{n}y_{i}x_{i}^{T}\beta +\left(1-y_{i}\right)\log\left(1+e^{z_{i}^{T}\theta }+e^{x_{i}^{T}\beta +z_{i}^{T}\theta }\right)\\ \; & -\sum_{i=1}^{n}\log\left(1+e^{x_{i}^{T}\beta }\right)-\sum_{i=1}^{n}\log\left(1+e^{z_{i}^{T}\theta }\right). \end{array}$$
To obtain reliable maximum likelihood estimators, it could be challenging to directly maximize the log-likelihood function in Equation (Equation 7). So, we suggest an EM algorithm to maximize the log-likelihood function.

Assume that an unobserved probability $w_{i}$ can determine whether $Y_{i}$ is a structural zero or not; when $w_{i}=1$ implies that $Y_{i}=0$ is a structural zero, and $w_{i}=0$ implies that the Bernoulli distribution determines the response $Y_{i}$ is 1 or 0. The likelihood function based on complete data D can be represented by
(8)$$\begin{array}{cc} L(\beta ,\theta |D)\; & =\prod_{i=1}^{n}f\left(w_{i}|z_{i}\right)f\left(y_{i}|w_{i},x_{i}\right)\\ \; & =\prod_{i=1}^{n}f\left(w_{i}=1|z_{i}\right)\prod_{i=1}^{n}f\left(w_{i}=0|z_{i}\right)f\left(y_{i}|w_{i}=0,x_{i}\right). \end{array}$$
The log-likelihood function can be represented by
(9)$$\begin{array}{c} L\left(\beta ,\theta |D,w\right)=\prod_{i=1}^{n}e^{w_{i}z_{i}^{T}\theta }{\left(1+e^{z_{i}^{T}\theta }\right)}^{-1}e^{y_{i}\left(1-w_{i}\right)x_{i}^{T}\beta }{\left(1+e^{z_{i}^{T}\theta }\right)}^{-(1-w_{i})}, \end{array}$$
and the log-likelihood function can be obtained by
(10)$$\begin{array}{c} \ell \left(\beta ,\theta |D,w\right)=\ell_{c}\left(\theta |w,z\right)+\ell_{c}\left(\beta |D,w\right), \end{array}$$
where
(11)$$\begin{array}{cc} \ell_{c}\left(\theta |w,z\right) & =\sum_{i=1}^{n}w_{i}z_{i}^{T}\theta -\log\left(1+e^{z_{i}^{T}\theta }\right) \end{array}$$
and
(12)$$\begin{array}{cc} \ell_{c}\left(\beta |D,w\right) & =\sum_{i=1}^{n}\left(1-w_{i}\right)\left\{y_{i}x_{i}^{T}\beta -\log\left(1+e^{x_{i}^{T}\beta }\right)\right\}. \end{array}$$

Using Equations (Equation 11) and (Equation 12) to implement the following E-step and M-step until convergences of the estimates of parameters, θ and β, are obtained.

E-step:Iteration (h+1): Estimate $w_{i}$ by its posterior mean $w_{i}^{\left(h+1\right)}$ given the estimates $\beta^{\left(h\right)}$ and $\theta^{\left(h\right)}$. If $y_{i}=1$, then $w_{i}^{\left(h+1\right)}=0$; otherwise,
(13)$$\begin{array}{cc} w_{i}^{\left(h+1\right)} & =\frac{P\left(Y_{i}=0|A\right)P\left(A\right)}{P\left(Y_{i}=0|A\right)P\left(A\right)+P\left(Y_{i}=0|A^{c}\right)P\left(A^{c}\right)} \end{array}$$
(14)$$\begin{array}{cc} & =\frac{1\times \delta_{i}}{1\times \delta_{i}+\left(1-p_{i}\right)\left(1-\delta_{i}\right)}=\frac{\delta_{i}}{1-p_{i}\left(1-\delta_{i}\right)}, \end{array}$$
where *A* and $A^{c}$ denote that $Y_{i}$ has a structure zero and is from a Bernoulli distribution.M-step:Maximizing
(15)$$\begin{array}{c} \ell_{c}\left(\theta |z,w^{\left(h+1\right)}\right)=\sum_{i=1}^{n}w_{i}^{\left(h+1\right)}z_{i}^{T}\theta^{\left(h\right)}-\log\left(1+e^{z_{i}^{T}\theta^{\left(h\right)}}\right) \end{array}$$
to obtain $\theta^{\left(h+1\right)}$, and maximizing
(16)$$\begin{array}{c} \ell_{c}\left(\beta |D,w^{\left(h+1\right)}\right)=\sum_{i=1}^{n}\left(1-w_{i}\right)\left\{y_{i}x_{i}^{T}\beta^{\left(h\right)}-\log\left(1+e^{x_{i}^{T}\beta^{\left(h\right)}}\right)\right\} \end{array}$$
to obtain $\beta^{\left(h+1\right)}$.

## 3. Examples

### 3.1. Example 1

A diabetes data set is used for illustration. The data set can be obtained from the R package version 2.1–3.1 “mlbench”. There are 768 records with eight independent variables including

The number of pregnancies.The glucose concentration in the 2-h oral glucose tolerance test.The diastolic blood pressure in mm Hg.The triceps skin-fold thickness in millimeters.Two-hour serum insulin in mu U/mL.Body mass.Diabetes pedigree function.Age in years.

The response variable is Diabetes, which can be negative or positive. After removing incomplete records, we have 392 records for the final sample for illustration. We labeled all the independent variables by $x_{i}$, i=1,2,⋯,8. Moreover, the variables of “The diastolic blood pressure in mm Hg”, “Body mass”, and “Age” are selected as the independent variables, labeled by $z_{h}$, h=1,2,3 to develop the second logistic regression model. The diabetes rate is 0.332 in this example.

All independent variables are scaling with $x_{ij}^{*}=\frac{x_{ij}-\min\left(x_{j}\right)}{\max\left(x_{j}\right)-\min\left(x_{j}\right)}$ and $z_{ih}^{*}=\frac{z_{ih}-\min\left(z_{h}\right)}{\max\left(z_{h}\right)-\min\left(z_{h}\right)}$ for i=1,2,⋯,n. Using the proposed EM algorithm, we can obtain the following ZIBer models,
(17)$$\begin{array}{cc} \log\left(\frac{p_{i}}{1-p_{i}}\right) & =-7.562+2.665x_{i1}^{*}+6.931x_{i2}^{*}+1.99x_{i3}^{*}-0.19x_{i4}^{*}+3.34x_{i5}^{*}\\ \; & +5.246x_{i6}^{*}+5.313x_{i7}^{*}-2.863x_{i8}^{*}, \end{array}$$
and
(18)$$\begin{array}{cc} \log\left(\frac{\delta_{i}}{1-\delta_{i}}\right) & =0.005+2.232z_{i1}^{*}-1.94z_{i2}^{*}-10.494z_{i3}^{*},\;i=1,\;2,\;\cdots ,\;392. \end{array}$$
When the estimated probability $p_{i}\geq 0.53$, i=1,2,⋯,392, the corresponding person is identified for diabetes. Based on the obtained model, the accuracy is 0.806. We want to study how much efficiency is lost using the typical logistic regression model. Based on the typical logistic regression model, we obtain the following estimation function form,
(19)$$\begin{array}{cc} \log\left(\frac{p_{i}}{1-p_{i}}\right) & =-5.771+1.397x_{i1}^{*}+5.434x_{i2}^{*}-0.122x_{i3}^{*}+0.628x_{i4}^{*}-0.687x_{i5}^{*}\\ \; & +3.449x_{i6}^{*}+2.664x_{i7}^{*}+2.037x_{i8}^{*},\;i=1,\;2,\;\cdots ,\;392. \end{array}$$
When the estimated probability $p_{i}\geq 0.39$, i=1,2,⋯,392, the corresponding person is identified for diabetes. Based on the obtained model, the accuracy is 0.798. The efficiency of the typical logistic regression model can be improved, but the improvement is not significant based on only 0.008 increment on the Accuracy. The diabetes rate is 0.332. If the success rate is high, the improvement of replacing the typical logistic regression model with the ZIBer regression model could not be significant. The finding makes sense because the imbalance in this example is insignificant. The R codes using the proposed EM algorithm to obtain the MLEs of the ZIBer regression model parameters are given in Appendix A. We use the best cut-rate of 0.53 to predict the value of Diabetes and identify it as positive if $p_{i}\geq 0.53$ and 0 otherwise. We can obtain the Accuracy, Sensitivity, and Specificity as 0.806, 0.608, and 0.905, respectively.

### 3.2. Example 2

A Taiwan credit data set is used for illustration. This data set can be obtained from the UC Irvine Machine Learning Repository by the hyperlink https://archive.ics.uci.edu/dataset/350/default+of+credit+card+clients; this data set was donated to UC Irvine Machine Learning Reponsitory on 25 January 2016. The response variable is the default payment, Yes = 1 for default and Yes = 0 for non-default. There are 30,000 records with the following 23 explanatory variables:The Given Credit Amount of NT dollars, including the individual consumer and his (or her) family (supplementary) credits.Gender, 1 for male and 2 for female.Education, 1 for graduate school, 2 for university, 3 for high school, and 4 for other.Marital Status, 1 for married, 2 for single, and 3 for other.Age in years.The history of past payments in 2005 from September to April in decent order of the columns. The measurement scale is −1 for pay duly, 1 for a one-month payment delay, 2 for a two-month payment delay, …, 8 for an eight-month payment delay, and 9 for a nine-month payment delay and above.The number of bill statements in 2005 with NT dollars from September to April in decent order of the columns.The amount of previous payments in 2005 with NT dollars from September to April in decent order of the columns.

The response variable of Default Payment is labeled by *Y*. We use the columns of The Given Credit Amount of NT dollars, Gender, Education, Marital Status, Age, and the payment statuses of April, May, June, July, and August as the independent variables, labeled by $X_{1}$, $X_{2}$, …, and $X_{10}$, respectively, to develop the ZIBer regression model. In many instances, we need to subjectively determine the independent variables of $Z^{\prime }s$ for establishing the second logistic regression model. In this example, we select the columns of Gender, Education, Marital Status, and Age as the independent variables of $Z^{\prime }s$ for the second logistic regression model.

No missing records were found in this data set. First, we searched the rows with labels that were not well-defined in the independent variables and removed all records that were not well-defined from the data set. Finally, 4030 records are used in the illustrative data set for constructing the ZIBer regression model.

The summary of the credit card users in the categories of Default Payment, Gender, Education, and Marriage is given below.

1436 default and 2594 non-default credit card users are included in the illustrative data set. The proportion is 35.63%.2385 female and 1645 male credit card users are in the illustrative data set. The proportions are 59.18% and 40.82%, respectively.The number of credit card users in the education categories of Senior High School, College, Graduate, and Others are 624, 1713, 1683, and 10, respectively. The proportions are 15.48%, 42.51%, 41.76%, and 0.25%.2076 credit card users are married, 1920 credit card users are single, and 34 credit card users are other. The proportions are 51.51%, 47.64%, and 0.85%.

Before model construction, we transform the categorical variable with three or larger categories into dummy variables. Hence, the Marital Status needs to be transformed into two dummy variables. If $X_{4}=1,2$, and 3, the new dummy variables of M1 and M2 can be obtained by $\left(M_{1},M_{2}\right)=\left(1,0\right),\left(0,1\right)$ and (1,1), respectively. Let $Z_{1}=X_{2}$, $Z_{2}=X_{3}$, $Z_{3}=M_{1}$, $Z_{4}=M_{2}$, and $Z_{5}=X_{5}$. All independent variables are scaling with $x_{ij}^{*}=\frac{x_{ij}-\min\left(x_{j}\right)}{\max\left(x_{j}\right)-\min\left(x_{j}\right)}$ and $z_{ih}^{*}=\frac{z_{ih}-\min\left(z_{h}\right)}{\max\left(z_{h}\right)-\min\left(z_{h}\right)}$ for i=1,2,⋯,n. Using the proposed EM algorithm, we can obtain the following ZIBer models,
(20)$$\begin{array}{cc} \log\left(\frac{p_{i}}{1-p_{i}}\right) & =0.639-1.266x_{i1}^{*}-0.458x_{i2}^{*}+0.554x_{i3}^{*}-3.713M_{i1}^{*}-3.039M_{i2}^{*}\\ \; & +1.789x_{i5}^{*}+0.179x_{i6}^{*}+4.520x_{i7}^{*}+6.366x_{i8}^{*}-0.908x_{i9}^{*}+3.876x_{i10}^{*}, \end{array}$$
and
(21)$$\begin{array}{cc} \log\left(\frac{\delta_{i}}{1-\delta_{i}}\right) & =4.860-0.441z_{i1}^{*}+0.024z_{i2}^{*}-3.970z_{i3}^{*}-3.754z_{i4}^{*}-0.495z_{i5}^{*},\\ \; & i=1,2,\;\cdots ,4030. \end{array}$$

When the estimated probability $p_{i}\geq 0.41$, i=1,2,⋯,4030, the corresponding customer is identified as a default. Based on the obtained model, the Accuracy is 0.911.

## 4. Monte Carlo Simulations

In this section, we would like to study the performance of the proposed ZIBer regression method and compare its performance with that of the ANN and LightGBM methods. To help readers have a comprehensive understanding of the Monte Carlo simulation design, the design of the Monte Carlo simulation section is explained in Section 4.1, and the implementation of performance comparison among models can be found in Section 4.2.

### 4.1. The Simulation Design

Before implementing Monte Carlo simulations, we need to generate imbalanced data. Hence, we need a baseline model to do a fair performance comparison. First, we generate zero-inflated samples. Assume that $w_{1},w_{2},\;\cdots ,w_{n}$ is a random sample from the uniform distribution over 0 and 1. if $w_{i}\leq \delta_{i}$, let $Y_{i}=0$ and $Y_{i}$ follows a Bernoulli distribution with probability $\pi_{i}=\left(1-\delta_{i}\right)p_{i}$ for i=1,2,⋯,n. In the simulation studies, we use three covariates: first logistic model, $x_{i1}=1$, generate $x_{i2}$ from the standard normal distribution, and generate $x_{i3}$ from the Bernoulli distribution with probability 0.75. Let $\beta_{1}=-1$, $\beta_{2}=1$ and $\beta_{3}=-2$. Present the corresponding logistic model by
$$\begin{array}{c} \log\left(\frac{p_{i}}{1-p_{i}}\right)=\beta_{1}x_{i1}+\beta_{2}x_{i2}+\beta_{3}x_{i3}=\beta_{1}+\beta_{2}x_{i2}+\beta_{3}x_{i3},\;i=1,\;2,\;\cdots ,\;n. \end{array}$$
We can obtain the value of $p_{i}$ from the equation,
$$\begin{array}{c} p_{i}=\frac{e^{\beta_{1}+\beta_{2}x_{i2}+\beta_{3}x_{i3}}}{1+e^{\beta_{1}+\beta_{2}x_{i2}+\beta_{3}x_{i3}}},\;i=1,\;2,\;\cdots ,\;n. \end{array}$$

Using two $z^{\prime }s$ for the second logistic model, let $z_{i1}=1$ and generate $z_{i2}$ from the Weibull distribution with a shape parameter 3.6 and scale parameter 1 for i=1,2,⋯,n. Let $\theta_{1}=-1$, $\theta_{2}=2$. Present the corresponding logistic model by
$$\begin{array}{c} \log\left(\frac{\delta_{i}}{1-\delta_{i}}\right)=\theta_{1}z_{i1}+\theta_{2}z_{i2}=\theta_{1}+\theta_{2}z_{i2},\;i=1,\;2,\;\cdots ,\;n. \end{array}$$
we can obtain the value of $\delta_{i}$ from the equation,
$$\begin{array}{c} \delta_{i}=\frac{e^{\theta_{1}+\theta_{2}z_{i2}}}{1+e^{\theta_{1}+\theta_{2}z_{i2}}},\;i=1,\;2,\;\cdots ,\;n. \end{array}$$
Then, we can generate $Y_{i}$ from a Bernoulli distribution with probability $\pi_{i}=\left(1-\delta_{i}\right)p_{i}$, i=1,2,⋯,n. Repeat the above process *n* times to obtain a zero-inflated random sample D.

### 4.2. The Implementation of Performance Comparison

To guarantee that the sample size is large enough to make the machine learning method perform stable, we consider n=5000 in this simulation study. The ZIBer algorithm can reach the stop rule if 50 iterative runs are up or all the parameter differences between two adjacent iterations are less than $10^{-5}$. For the comparison purpose, we use 80% or 4000 generated data to train the LightGBM and ANN models and obtain the MLEs of ZIBer model parameters. R codes were prepared to implement the prediction performance comparison. When the MLEs of the model parameters are obtained, the model global searches for the best cut point with the lowest error classification rate as the training ZIBer model. Because the separation of training and testing groups depends on random chance, we repeated the whole process 1000 times and calculated the average performance of accuracy, sensitivity, and specificity,
(22)$$\begin{array}{cc} \mathrm{Accuracy} & =\frac{\mathrm{TP}+\mathrm{TN}}{\mathrm{TP}+\mathrm{TN}+\mathrm{FP}+\mathrm{FN}}, \end{array}$$
(23)$$\begin{array}{cc} \mathrm{Sensitivity} & =\frac{\mathrm{TP}}{\mathrm{TP}+\mathrm{FN}}, \end{array}$$
(24)$$\begin{array}{cc} \mathrm{Specificity}= & \frac{\mathrm{TN}}{\mathrm{TN}+\mathrm{FP}}, \end{array}$$
which TP denotes true positive, FN denotes false negative, FP denotes false positive, and TN denotes true negative. The simulation results are given in Table 1. The dispersion of the metrics Accuracy, Sensitivity, and Specificity of 1000 repetitions is given in Figures 3–5.

Table 2 shows that no method dominates the others. These three methods have similar performance for Accuracy and Specificity. The differences between these three methods are almost ignored. However, we found that the ANN method has extreme results for Sensitivity and Specificity. The average sensitivity based on 1000 repetitions is nearly 0, and the average Specificity based on 1000 repetitions is 1. The finding means the ANN methods could not be suitable for the case, emphasizing the Sensitivity measure. The Sensitivity in zero-inflated-imbalanced data is not easy to measure correctly. The LightGBM and ZIBer methods outperform the ANN methods in Sensitivity and Specificity measures. We also find that the ZIBer and LighGBM methods are competitive.

From Figure 1, Figure 2 and Figure 3, we can see the spread of 1000 measures of Accuracy, Sensitivity, and Specificity, respectively. Figure 1 indicates the Accuracy of LightGBM has a higher median than the other two competitors with a shorter dispersion than the ZIBer method. Figure 2 shows the strength of the ZIBer method with a higher median than the other two competitors and a similar dispersion with the LightGBM method. Figure 3 shows the ZIBer method has the lowest median and is widespread compared with the other two competitors.

Let us revisit Examples 1 and 2, these two data sets can be zero-inflated because the proportions of non-diabetes and non-default payments are 66.8% and 64.4%, respectively. Because the simulation design used is the same and comparison results are similar, Example 2 will be kept for explanation. In practical applications, some non-default credit card users have extremely low default probability. They could have a different model structure from users that could be default but not default in the current state. The performance of the ANN, LightGBM, and ZIBer models is compared using model validation. Randomly splitting the illustrative data set into the training data set with 80% observations, and the remaining 20% observations comprise the testing data set. Meanwhile, the random sampling maintains the same proportion of default payments in the training data set as the illustrative data set. The training data set is used to train the ANN and LightGBM models and construct the ZIBer model. Then, the testing data set is used to evaluate the Accuracy, Sensitivity, and Specificity. The split operation is repeated 300 times. Accuracy, Sensitivity, and Specificity are evaluated using their mean values in 300 repetitions. Moreover, the standard deviations of these three models in 300 repetitions are computed, too.

All computation results are given in Table 2, which shows that these three models are competitive. No one model dominates the others. The LightGBM model has the highest Accuracy and Specificity. The ZIBer model has the highest Sensitivity. We also note that the ZIBer model and LightGBM are competitive for the Accuracy measurement. The mean Sensitivity of ANN is significantly lower than the other two competitors and has the largest standard deviation. We also find that the standard deviation of the Sensitivity of LightGBM is over two times higher than that of the ZIBer model.

Figure 4 and Figure 5 show the scatter plots of 300 Sensitivity measurements versus the Accuracy measurements and 300 Sensitivity measurements versus the Specificity measurements for the ANN, LightGBM, and ZIBer models. These two scatter plots show that the Accuracy, Sensitivity, and Specificity measures of the ZIBer model spread at the top position among all three methods. In this example, the ZIBer model looks to have a more stable performance than its competitors.

The ZIBer method is a statistical model-based method. Hence, the ZIBer method does not require big data to train models. When the sample size is small, the ZIBer method can be expected to perform best. The ZIBer and LightGBM methods are more competitive than the ANN, and we recommend using them to process zero-inflated-imbalanced data.

## 5. Conclusions

This paper proposes the EM maximum likelihood estimation procedure to obtain reliable MLEs of the ZIBer model parameters. The EM algorithm is simple for practical use. The machine learning models of ANN and LightGBM are used for comparing the performance of the proposed ZIBer model in terms of Accuracy, Sensitivity, and Specificity. Monte Carlo Simulations were conducted, and the simulation results show that no method can dominate the other two methods under the zero-inflated-imbalanced data.

Two examples of diabetes and credit card defaults are used for illustration. We demonstrate how to use the proposed EM algorithm to obtain the reliable parameters of the ZIBer regression model. Moreover, we use a random sampling technique for Example 2 to study the performance of the ANN, LightGBM, and ZIBer models under zero-inflated-imbalanced binary data sets in Section 4. We find that the LightGBM and ZIBer models are competitive. The LightGBM has the highest Accuracy and Specificity, and the ZIBer model has the highest Sensitivity. The ANN model performs worse in the measure of Sensitivity than the two competitors. Overall, the ZIBer model has a more stable performance than its competitors.

The strength of a parametric statistical model is the functional form is clear, and the explanation between the response and explanatory variables is feasible. However, screening out critical explanatory variables from a large set in the model is challenging. Enhancing the abilities of the model with higher sensitivity of the ZIBer model is also challenged. These two issues could be studied in the future. 

## Figures and Tables

**Figure 1 entropy-26-00015-f001:**
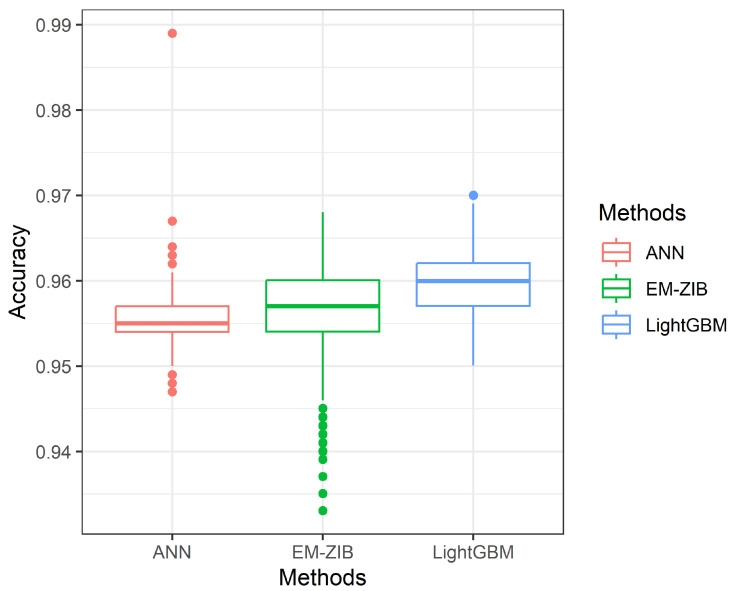
The box plots of accuracy for the ANN, LightGBM, and ZIBer models.

**Figure 2 entropy-26-00015-f002:**
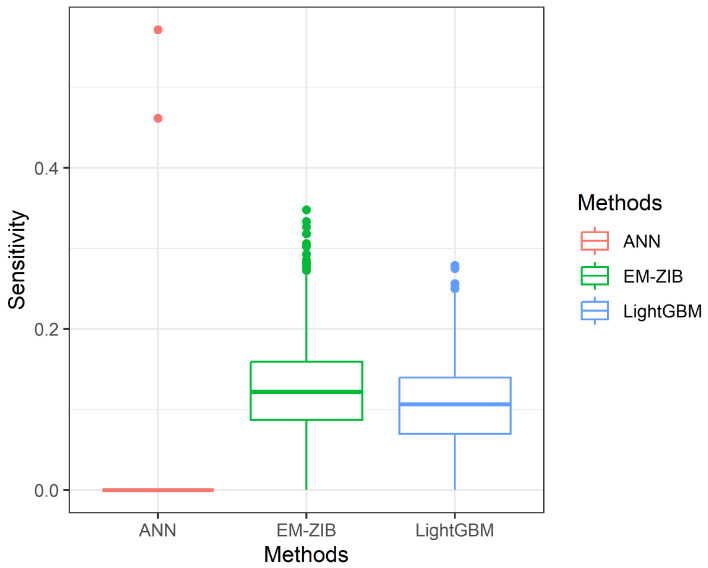
The box plots of sensitivity for the ANN, LightGBM, and ZIBer models.

**Figure 3 entropy-26-00015-f003:**
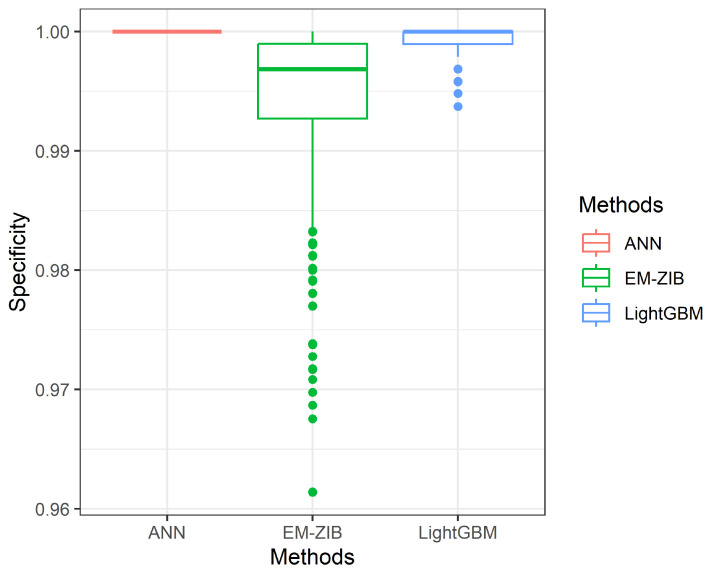
The box plots of Specificity for the ANN, LightGBM, and ZIBer models.

**Figure 4 entropy-26-00015-f004:**
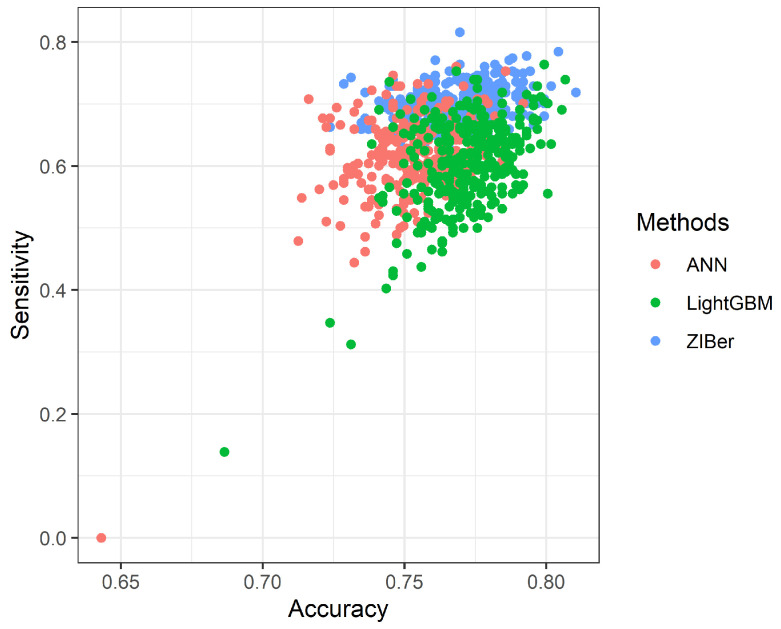
The scatter plot of the Accuracy versus Sensitivity for the ANN, LightGBM, and ZIBer models.

**Figure 5 entropy-26-00015-f005:**
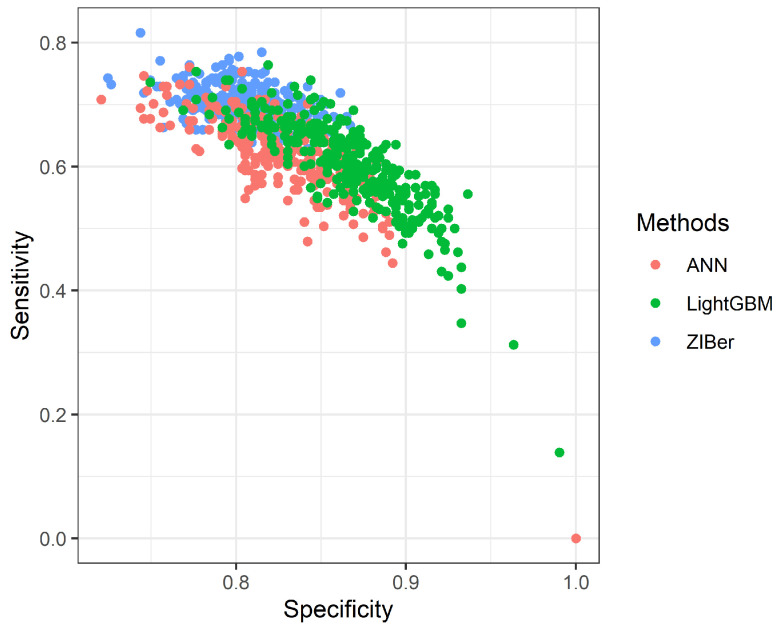
The scatter plot of the Specificity versus Sensitivity for the ANN, LightGBM, and ZIBer models.

**Table 1 entropy-26-00015-t001:** Performance comparison among ZIBer, LightGBM, and ANN based on 1000 repetitions.

	Average Values		
**Methods**	**Accuracy**	**Sensitivity**	**Specificity**
ANN	0.9554	0.0010	1
LightGBM	0.9596	0.1082	0.9994
ZIBer	0.9563	0.1256	0.9952

**Table 2 entropy-26-00015-t002:** Performance comparison among the ANN, LightGBM, and ZIBer models based on 100 split operations for the illustrative data set.

	Mean Values (Standard Deviation*100)		
**Methods**	**Accuracy**	**Sensitivity**	**Specificity**
ANN	0.7402 (3.9677)	0.5400 (21.2622)	0.8513 (6.5206)
LightGBM	0.7718 (1.4739)	0.6042 (7.4150)	0.8647 (3.6120)
ZIBer	0.7692 (1.4731)	0.7020 (3.0221)	0.8064 (2.4367)

## Data Availability

The research data can be free access by checking https://archive.ics.uci.edu/dataset/350/default+of+credit+card+clients, this data set was donated to UC Irvine Machine Learning Repository on 25 January 2016.

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
