# Peer review of "Binary Classification with Imbalanced Data"

_entropy, 2023, doi:10.3390/e26010015_

Round 1
Reviewer 1 Report
Comments and Suggestions for Authors
The authors dealt with the well-known in the literature problem of classification with poorly balanced classes of the dependent variable. They present an original version of a classifier using the Zero-Inflated Bernoulli Regression Model. However, while the problem itself may be known, each new solution sheds new light on the matter and is interesting in this context. The authors present a rich literature review and logically provide the motivation for their solution. The ZIBer preformance results are compared with the ANN and LightGBM algorithms on the well-known UC Irvine Machine Learning Repository : https://archive.ics. 377 uci.edu/dataset/350/default+of+credit+card+clients. The results confirm the importance of this approach.
Author Response
Thank you for your constructive comments and suggestions for our study. We follow all the comments and suggestions to improve our manuscript. You can find the changes in the revision from the point-to-point responses that are submitted to this submission.

Reviewer 2 Report
Comments and Suggestions for Authors
in this paper, the authors tackled an important problem of training machine learning models from imbalanced datasets. The topic is certainly worthy of investigation, as imbalanced datasets are intrinsically captured for lots of real-world classification problems. The manuscript, however, suffers from quite a number of shortcomings which should be, in my opinion, thoroughly addressed before it could be considered for publication:
1. Please avoid bulked referencing (see e.g., line 52), as it does not bring any useful information to the reader.
2. The authors should thoroughly rework the related literature part of the manuscript, as it is currently fairly shallow (see e.g., the work by Krawczyk to see what are the approaches in the state of the art to tackle the imbalanced data problems: https://link.springer.com/article/10.1007/s13748-016-0094-0; also, there are a multitude of approaches that are used at both algorithm- and data-level to compensate for such difficulties, see e.g., https://link.springer.com/article/10.1007/s10462-017-9611-1).
3. Instead of enumerating selected approaches from the state of the art, the authors should focus on more critical analysis of the state of the art. It would be useful to concisely present (perhaps in a tabular form) the strengths and weaknesses of the existing approaches, so that it is clear which open research gaps are being addressed in this paper. This will also help better highlight the most important contributions behind the work reported here.
4. I would suggest avoiding paragraphs which contain a single sentence only (see e.g., line 177).
5. It may be useful to provide a bit more abstract description in Section 2.1 – is it necessary to say about “the infection status” in the introductory part of this paragraph? Perhaps it may be easier to keep this description at a higher abstraction level, as the algorithms are applicable to various downstream tasks.
6. The authors mix conceptualization with implementation, e.g., in Section 3 – is it necessary to report implementation details here? I suggest moving the discussion on the R packages and implementation to the experimental section.
7. The manuscript should be restructured – please provide an Experimental Study section, and split Section 3 into the conceptualization (simulation design) and experiments. Also, Section “An example” could fit there. Similarly, it is unclear why did the authors decide to include the description of the quality metrics here. Overall, the structure of the manuscript should be thoroughly revised.
8. The quality of the figures must be improved – all of them should be in a vector format (they should be high-resolution).
9. We are currently facing the reproducibility crisis in the machine learning research – to this end, the authors should make their implementation publicly available so that other research groups can easily reproduce the experiments.
10. The experimental study is very limited, thus it is not very convincing. Please increase the number of benchmark datasets taken for analysis (also, please include some extremely imbalanced datases, with e.g., imbalance ratio IR>20), and please back up the analysis with appropriate statistical testing (in other words, are the differences across the investigated algorithms statistically significant? Please report appropriate p-values).
Comments on the Quality of English LanguageThe English requires refinements.
Author Response

(The authors gave the same response as above.)

Round 2
Reviewer 2 Report
Comments and Suggestions for Authors
Thank you indeed for addressing most of my concerns. I do still, however, encourage the authors to expand the experimental part of the manuscript, and to include more datasets that would show the generalization capabilities of the proposed framework.
Comments on the Quality of English LanguageOverall, the English is fine.
Author Response
We want to express our deep thanks for the comment of Reviewer 2. The changes in Revision 2 are summarized as follows:
- We cut Equations (8) short for a clear expression.
- The typos in Equations (18) and (21) were fixed.
- Including a third or more examples cannot make the length of the paper concise for reading. Moreover, the experimental studies will be similar. To help users have a comprehensive understanding of the use of the proposed method. We add the R codes using the proposed EM algorithm to obtain the MLEs of the ZIBer regression model parameters given in Appendix A. We use the best cut-rate of 0.53 to predict the value of Diabetes and identify it as positive if p>= 0.53 and 0 otherwise. We can obtain the Accuracy, Sensitivity, and Specificity as 0.806, 0.608, and 0.905, respectively. Please see the paragraph under Equation (19).
We also add the following paragraph in lines 342 to 345 to explain the reason for only including Example 2 in the experimental design:
“Let us revisit Examples 1 and 2, these two data sets can be zero-inflated because the proportions of non-diabetes and non-default payments are 66.8\% and 64.4\%, respectively. Because the simulation design used is the same and comparison results are similar. Example 2 will be kept for explanation.”